# Constructing a draft Indian cattle pangenome using short-read sequencing
Sarwar Azam [1,2], Abhisek Sahu[1], Naveen Kumar Pandey [1], Mahesh Neupane[3], Curtis P. Van Tassell [3], Benjamin D. Rosen [3 ✉], Ravi Kumar Gandham[1], Subha Narayan Rath[2] & Subeer S. Majumdar [1 ✉]

Indian *desi* cattle, known for their adaptability and phenotypic diversity, represent a valuable genetic resource. However, a single reference genome often fails to capture the full extent of their genetic variation. To address this, we construct a pangenome for *desi* cattle by identifying and characterizing non-reference novel sequences (NRNS). We sequence 68 genomes from seven breeds, generating 48.35 billion short reads. Using the PanGenome Analysis (PanGA) pipeline, we identify 13,065 NRNS (~41 Mbp), with substantial variation across the population. Most NRNS were unique to *desi* cattle, with minimal overlap (4.1%) with the Chinese indicine pangenome. Approximately 40% of NRNS exhibited ancestral origins within the Bos genus and were enriched in genic regions, suggesting functional roles. These sequences are linked to quantitative trait loci for traits such as milk production. The pangenome approach enhances read mapping accuracy, reduces spurious single nucleotide polymorphism calls, and uncovers novel genetic variants, offering a deeper understanding of *desi* cattle genomics.

The first pangenome was developed by Tettelin et al. in the early 2000s for *Staphylococcus agalactiae*, highlighting the limitations of single reference genomes for capturing intraspecies genetic diversity[1]. Pangenomes are constructed by sequencing multiple genomes of individuals within a species, providing a more comprehensive view of genetic variation. While initially focused on prokaryotes[2,3], plants[4,5], and fungi[6], advancements in next-generation sequencing (NGS) technology, coupled with more affordable sequencing costs and accessible computational resources, have facilitated pangenome construction for animals with large genomes. Efforts to establish a human pangenome[7–11] culminated in remarkable studies establishing human pangenomes of African[12] and Han Chinese[13] descent. Similar progress has been made with livestock species like goats[14], sheep[15], and pigs[16,17]. Recently, pangenomes for cattle of European[18] and Chinese origin[19,20] have also been published, marking a major advancement in that field.

Cattle (*Bos taurus*) encompass two distinct subspecies: *Bos taurus taurus* and *Bos taurus indicus*, originating from separate domestication events[21,22]. *Bos taurus indicus* commonly referred as *Bos indicus*, also known as zebu or *desi* cattle, were domesticated in the Indus Valley Civilization and are prevalent in the Indian subcontinent[23]. India, with 13.1% of the world's cattle population[24], is home to numerous *desi* breeds notable for their adaptation to harsh environments, including resistance to tropical diseases and pests and the ability to prosper while eating low-quality forage. The State of the World's Animal Genetic Resources (SoW-AnGR) identifies 60

local, 8 regional transboundary, and 7 international transboundary cattle breeds originating in India[25]. These groups serve diverse purposes, with some breeds, like Sahiwal, Gir, Rathi, Tharparkar, and Red Sindhi primarily maintained for milk production, while others like Deoni, Hariana, and Kankrej are dual-purpose or draft breeds[26,27]. Therefore, a single reference genome assembly may not adequately capture the extensive genetic diversity present among Indian cattle breeds.

Pangenomes provide a comprehensive view of genetic variation by capturing insertion sequences missing from the reference genome[12,28], termed as non-reference novel sequences (NRNS) in this study. The completeness of the pangenome relies on the availability of genome data, with a large number of de novo genome assemblies resulting in a high quality pangenome. Pangenomes can be constructed using both long and short-read NGS data[17]. Long reads offer superior de novo assemblies, but generating them for species with large genomes, like cattle, remains expensive and time consuming[29]. An alternative approach, widely adopted in many pangenome studies, involves constructing pangenomes from short reads. Short-read-based pangenome studies have identified substantial NRNS in humans (10 Mb in African population[12], 15 Mb in Han Chinese population[13], goats (38.3 Mb)[14] and pigs (132.4 Mb)[30]. Similarly, 83 Mb of NRNS in European cattle (*Bos taurus*) was identified using a short-read approach by Zhou et al.[20]. Recent efforts in cattle have expanded beyond single-species pangenomes. Notably, studies by Leonard et al.[31] and

[1]National Institute of Animal Biotechnology, Hyderabad, India. [2]Indian Institute of Technology Hyderabad, Sangareddy, India. [3]Animal Genomics and Improvement Laboratory, USDA-ARS, Beltsville, MD, USA. ✉e-mail: ben.rosen@usda.gov; subeer@niab.org.in

Crysnanto et al.[18] utilized long reads to construct pangenomes representing the entire Bos genus, incorporating *Bos taurus* and wild cattle relatives. This approach offers a broader view of genetic variation within the Bos lineage, but focuses on multi-species comparisons rather than the detailed exploration of within-species diversity within *Bos indicus*.

Building upon these advancements, this study aimed to construct an intra-species *desi* cattle pangenome specifically focusing on *desi* cattle breeds from India. We sequenced the genomes of 67 individuals representing 7 different Indian breeds using short-read NGS and mapped them to the *Bos indicus* reference genome to identify NRNS. We further characterized these NRNS to understand their functional potential and variation within the *desi* cattle population. Additionally, we validated the constructed pangenome by aligning sequencing data from genomes of additional individuals representing diverse populations. This analysis demonstrated the pangenome's superiority over the reference genome in terms of read mapping efficiency and its ability to capture novel single nucleotide polymorphisms (SNPs).

## Results

### *Desi* cattle genome sequencing

We conducted whole-genome sequencing on 68 *desi* cattle individuals representing 7 distinct breeds. This effort generated a total of 48.35 billion reads (Supplementary Table 1). Following rigorous preprocessing procedures to eliminate low-quality reads, we retained 43.79 billion high-quality reads. This yielded a dataset of ~6062 Gb of sequence data, thereby achieving an average coverage depth of approximately 33X for the cattle, based on an estimated genome size of approximately 2.7 Gb.

### NRNS identification using Pangenome Analysis (PanGA) pipeline

The PanGA pipeline is designed to simplify various processes involved in identifying NRNS for constructing a short-read-based pangenome. This comprehensive pipeline encompasses preprocessing of raw reads, filtration of low-quality reads, alignment to a reference genome, extraction of unaligned reads, and subsequent assembly into contigs (Fig. 1). Additionally, it integrates procedures to identify and eliminate contaminant sequences from de novo assembled contigs. The resulting contigs are merged to form a non-redundant set of sequences, designated as NRNS. We executed this pipeline on the preprocessed data from all 68 samples, totaling 21,893 million paired-end reads. These reads were aligned to the Brahman reference genome (GCF_003369695.1), achieving an overall average alignment percentage ranging from a minimum of 95.86% to a maximum of 99.24%

(Supplementary Table 2). Subsequently, the pipeline gathered a cumulative total of 1073 million paired-end unaligned reads and 649 million single-end reads from all samples. The pipeline then performed separate de novo assemblies for each animal using their corresponding unaligned reads, resulting in a collective total of 1,152,124 contigs (Supplementary Table 3). When we selected contigs longer than 1000 bp in every sample, we obtained 244,355 contigs for further analysis.

These contigs underwent contaminant screening using our internally developed Fasta2lineage tool (Supplementary Fig. 1), which was integrated into our PanGA pipeline. Following this screening, a cumulative total of 50,229 contaminant sequences, primarily consisting of environmental sequences (61.48%), known bovine parasites including *Theileria annulata* (24.72%), *Theileria orientalis* (7.87%), and *Babesia bigemina* (1.22%), and other minor contaminants below 1%, were removed from the sequence data (Supplementary Table 4), leaving 194,126 de novo contigs for subsequent processing (Supplementary Table 3). These de novo contigs, totaling approximately 538 Mb, were then merged to create a non-redundant set of contigs. As a result, PanGA pipeline generated 13,065 sequences designated as NRNS from the cattle population, with a combined length of 40,973,925 bp (Supplementary Data 1). The sizes of these NRNS ranged from 1000 to 58,737 bp, with an N50 of 4629 bp (Fig. 2B).

### Comparison with published pangenomes and other genome assemblies

When aligned with insertion sequences identified in the pan-genome study by Zhou et al.[20], we detected only 183 NRNS sequences spanning approximately 300 kb (Table 1). In contrast, when compared with the pan-genome study of 10 cattle representing 10 Chinese indicine breeds by Dai et al.[19], we identified 747 matching NRNS sequences spanning 1.68 Mb. The higher number of matches with Dai et al.'s study was expected, as both studies focused on indicine cattle. However, this match represents only about 4.1% of the 40.97 Mb NRNS identified in this study.

Furthermore, NRNS contigs were aligned with recently available haplotype-resolved genome assemblies of Tharparkar and Sahiwal breeds. Notably, 2466 contigs (9.75 Mb) and 2652 contigs (10.57 Mb) aligned specifically to the Tharparkar and Sahiwal assemblies, respectively. We detected 3512 contigs totaling 13.2 Mb (32%) aligning to either the Tharparkar or Sahiwal assembly. Further, we also performed alignment of NRNS contigs on the genetically distant genome assembly of the Chinese indicine Hainan breed[32], resulting in the alignment of 3596 contigs (12.87 Mb).

**Fig. 1 | PanGA pipeline for identification of NRNS in *Bos Indicus*.** The flowchart depicts the sequential steps involved in the PanGA pipeline for identifying NRNS in *Bos Indicus*. The pipeline outlines the tools and procedures employed to select and extract NRNS from the dataset.

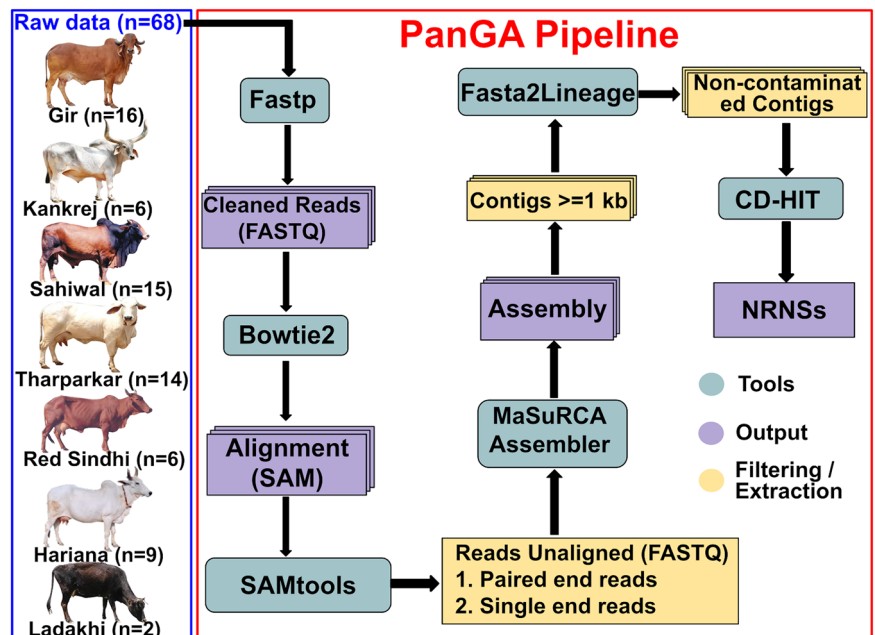

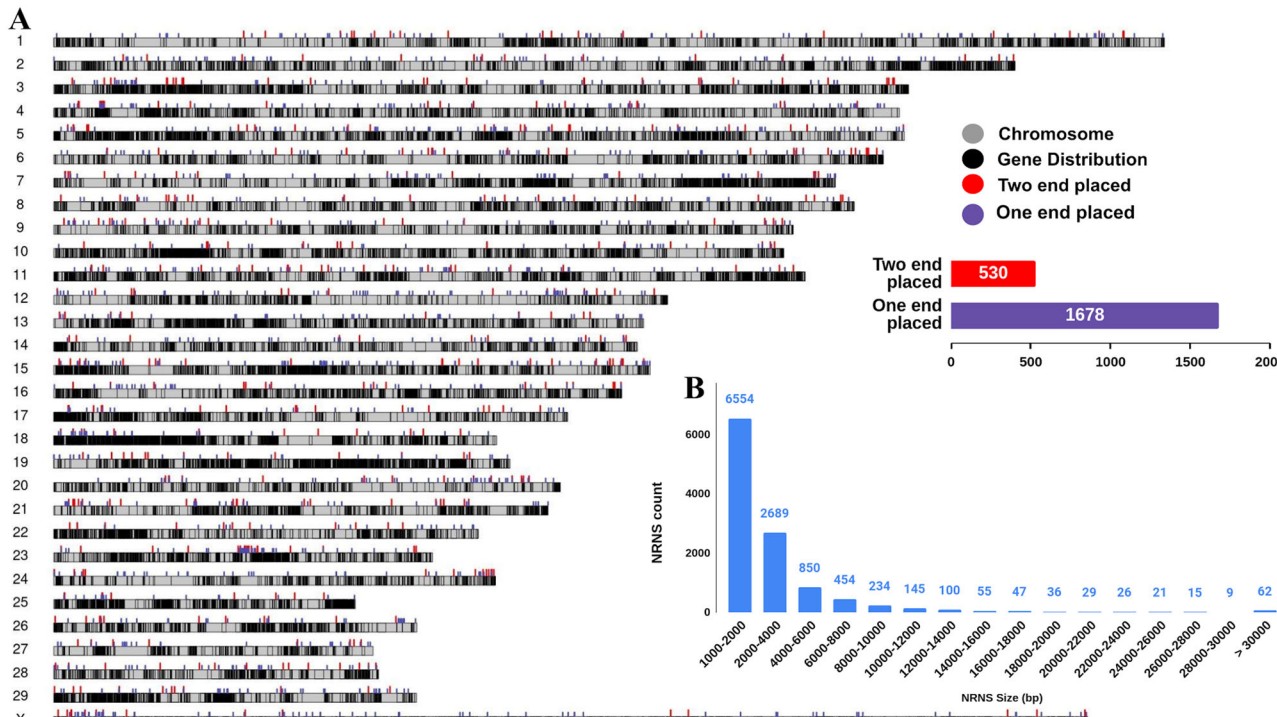

**Fig. 2 | Overview and size distribution of NRNS. A** Ideogram showing the distribution of NRNS on Brahman reference chromosomes. Violet and red tracks represent NRNS with one and two ends anchored to the reference genome, respectively. **B** Size distribution of NRNS. The barplot shows the distribution with a bin size of 2000 bp, with the first bin representing the size range of 1000–2000 bp due to the minimum NRNS size of 1000 bp.

### Table 1 | Comparison of NRNS with published pangenomes and other genome assemblies

| Category | Novel insertion in published pangenome | | Complete genome assembly | | |
|---|---|---|---|---|---|
| | **Zhou et al. 2022** | **Dai et al. 2023** | **Tharparkar** | **Sahiwal** | **Hainan** |
| Size | 94.04 Mb | 124.41 Mb | 2.663 Gb | 2.664 Gb | 2.616 Gb |
| Number of NRNS get hit | 183 | 747 | 2466 | 2652 | 3596 |
| Total aligned length | 299.49 Kb | 1.68 Mb | 9.75 Mb | 10.57 Mb | 12.87 Mb |
| Largest NRNS matched (bp) | 5341 | 15,288 | 50,343 | 58,737 | 58,737 |

Tharparkar (GCF_029378745.1), Sahiwal (GCA_029378735.1), Hainan (GCA_039881165.1)

### Table 2 | NRNS presence/absence and length distribution data statistics

| Category | Number of contigs | Mean number of insertions per individual | Mean number individuals per insertion (of 68) | Total size (bp) | Largest Contig |
|---|---|---|---|---|---|
| Two ends placed | 536 | 140 (26.12%) | 18 | 891,174 | 7171 |
| One end placed | 1766 | 412 (23.33%) | 16 | 4,512,132 | 30,850 |
| Unplaced | 10,763 | 1748 (16.24%) | 11 | 35,570,619 | 58,737 |
| Total | 13,065 | 2300 (17.60%) | 12 | 40,973,925 | 58,737 |
| Non-private only | 8402 | 2232 (26.57%) | 18 | 33,808,020 | 58,737 |
| Private only | 4663 | 68 (0.01%) | 1 | 7,165,905 | 21,935 |

### Calling presence or absence of NRNS per sample

Once the NRNS in the samples were identified, we assessed their presence and absence in each of the *desi* samples. Out of the 13,065 NRNS identified, 8402 sequences, spanning a combined length of 33,808,020 bases, were found in at least two individuals within the *desi* sample cohort. However, 4663 NRNS, comprising 35.70% of the total NRNS, were found in just one individual and considered as singletons or private insertions. On average,

each NRNS was detected in 12 out of 68 individuals, whereas this number increased to 18 when considering non-private NRNS only (Table 2). Conversely, each individual represented an average of 2300 NRNS in the cohort. Further analysis of NRNS presence/absence by breed revealed that 2317 NRNS were present in all the breeds (Supplementary Fig. 2). Additionally, there were breed-specific NRNS, with the highest number found in Gir (1400) and the lowest in Ladakhi (264). However, the limited sample size

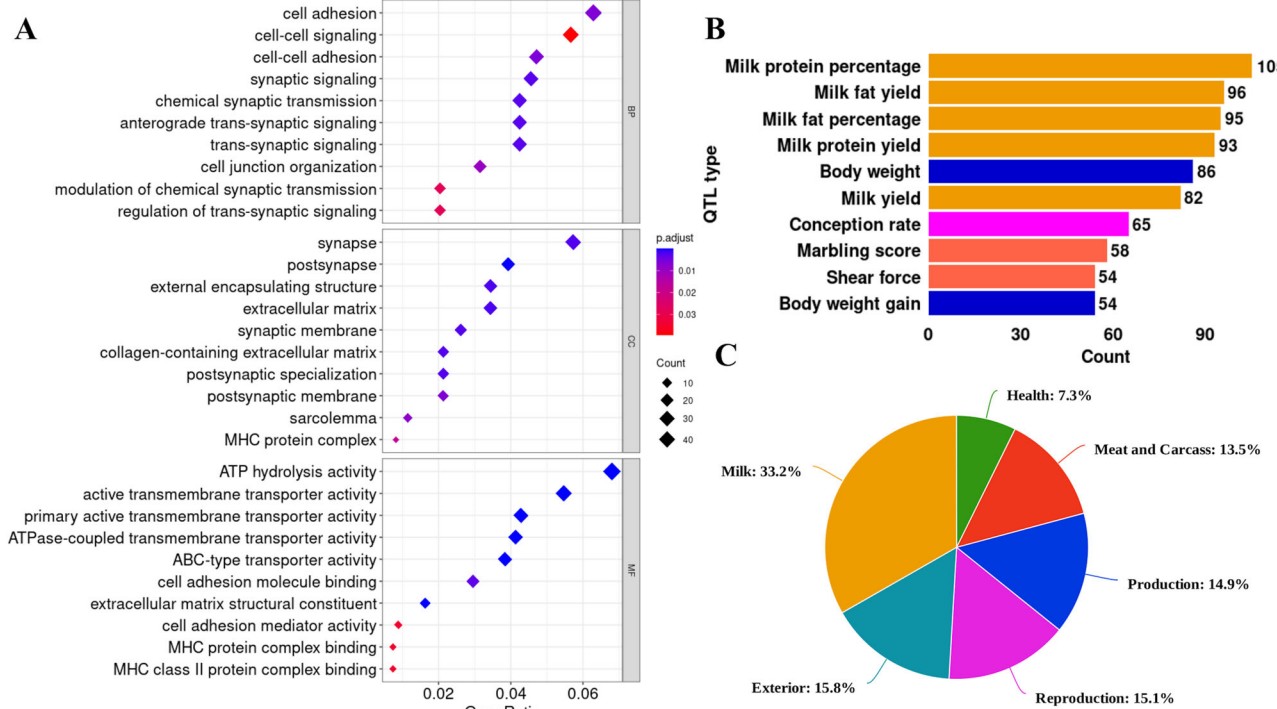

**Fig. 3 | Characterization of transcriptome in NRNS. A** Gene Ontology (GO) enrichment analysis plot for the biological processes (BP), molecular function (MF), and cellular component (CC) associated with NRNS. **B** Top ten enriched QTL type (bar plots) associated with NRNS. **C** Percentage of QTL trait type (pie chart) associated with NRNS.

per breed might influence these observations. A larger specific sample size would be necessary to definitively characterize the breed specific NRNS pattern.

### Placement of NRNSs onto the reference genome

We mapped NRNS to the reference genome by linking information from their ends to chromosomes using evidence from anchored reads. Out of 13,065 NRNS contigs, 2302 (17.61%) were successfully placed on chromosomes (Table 2; Supplementary Data 2). The majority of the NRNS contigs, 10,670 (~81.6%), were not assigned to chromosomes. Additionally, a small fraction, 93 contigs (0.7%), were discarded due to multiple mappings or inconsistencies in mapping direction. Of these mapped NRNS, 536 were fully resolved with both ends placed on chromosomes (Supplementary Data 3) and remaining 1766 NRNS exhibited partially resolved with only one end mapped (Supplementary Data 4; Fig. 2A). Among all the placed NRNS, 1015 were located within 679 genes, including 31 lncRNA genes, 630 protein-coding genes, and 18 pseudogenes. Among the 630 protein-coding genes, NRNS overlapped with 90 exons, whereas the remaining NRNS were located within introns. The largest fully resolved NRNS spanned 7171 bp and was present in 50 individuals, while the largest unplaced NRNS was 58,737 bp long and appeared in 33 samples. Notably, the average number of individuals per insertion was 18 for both ends placed NRNS, 16 for one end placed NRNS, and 11 for unplaced NRNS. This observation shows that placed NRNS were genotyped in a higher number of individuals compared to unplaced NRNS.

To further investigate the factors influencing NRNS placement, we analyzed the impact of Tandem Repeats (TRs) on mapping success. Annotation of TRs revealed that 2823 out of 13,065 NRNS contained TRs with scores >100. As expected, mapping of NRNS containing TRs proved more challenging. Out of 2823 TR-containing contigs, only 328 were placed on chromosomes (11.61%), including 212 single-end and 116 paired-end contigs. This lower placement rate for TR-containing NRNS compared to the overall placement rate of 17.61% highlights the impact of TRs on the successful mapping of NRNS.

### Transcription potential of NRNS

To identify the transcriptional potential of the NRNS, we predicted the transcript structure using both ab initio and evidence-based methods. A total of 1856 peptides were predicted using Augustus, while 7991 transcripts were predicted using StringTie. These peptides or transcripts in NRNS sequences can be part of larger proteins. To find such proteins, these transcripts were matched with close relatives and further with the Chordata protein database. This provided a total of 880 complete known genes. It was observed that many of the predicted transcripts partially matched a small number of proteins.

Additionally, we considered the previously identified 630 protein-coding genes in which NRNS are placed, suggesting they might modify gene structure in different individuals. We created a comprehensive set of genes by merging the 880 annotated genes in NRNS with the 630 genes in which NRNS were placed. This resulted in a total of 1453 non-redundant genes (Supplementary Table 5), which can be used for downstream functional analysis.

### Gene ontology (GO) enrichment analysis

To evaluate the potential impact of NRNS on biological functions, we performed GO enrichment analysis using the set of 1453 genes associated with NRNS. This analysis revealed significant enrichment ($P <= 0.05$) for GO terms associated with various biological processes, cellular components, and molecular functions (Fig. 3A). The most enriched GO terms for biological processes included cell adhesion, cell-cell signaling, and cell-cell adhesion. This suggests that NRNS may play a crucial role in cell communication and organization within tissues. GO terms associated with synapses and post synapses were significantly ($P <= 0.05$) enriched within the cellular component category. This finding implies that NRNS might be involved in neuronal function and communication at the synapse level. Terms related to ATP transporter activity and other transporter activities were the most enriched within the molecular function category. Additionally, GO terms related to MHC protein complex binding were significantly ($P <= 0.05$) enriched in the molecular function category. These

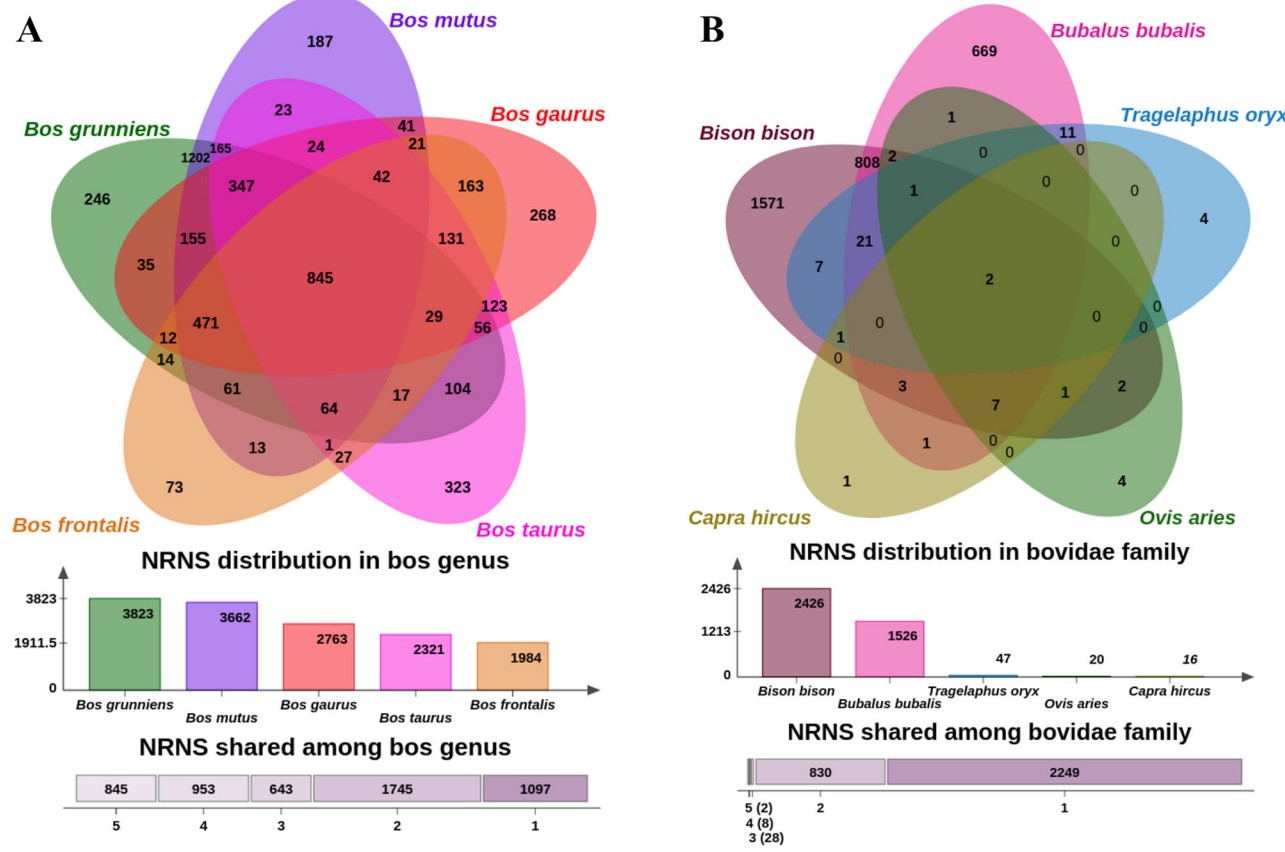

**Fig. 4 | Evolutionary conservation of NRNS. A** Venn diagram illustrating the distribution of NRNS across Bos species. Overlapping regions indicate shared NRNS among these closely related species. **B** Venn diagram comparing NRNS presence across the broader Bovidae family, revealing the extent of sequence conservation at a higher taxonomic level.

results suggest that NRNS-associated genes potentially influence the movement of molecules across membranes and also play a role in immune responses.

Further analysis of GO enrichment for genes from cloud-like and core-like NRNS contigs revealed distinct patterns (Supplementary Fig. 3). Genes associated with core-like NRNS contigs showed significant enrichment for GO terms related to MHC class II, antigen processing and presentation via MHC class II, *and* MHC protein complex assembly. This suggests that core-like NRNS genes may have important roles in immune functions. In contrast, genes from cloud-like NRNS contigs were significantly enriched for GO terms related to cellular component morphogenesis, indicating potential roles in cell structure and development.

### Quantitative trait locus (QTL) analysis

When the identified NRNS genes were analyzed for QTL associations, 500 genes were found to fall within important QTL regions. Altogether, these genes can be associated with a redundant set of 2773 QTLs, as one region may be associated with multiple QTLs. We further examined the top QTLs enriched with a higher number of genes and found that the most enriched QTL was milk protein percentage, associated with 105 genes (Fig. 3B). These QTLs were then categorized into six trait categories based on the cattle QTLdb: health, meat and carcass, milk production, reproduction, and exterior. The majority of QTLs were found to be related to milk traits (33.2%), while the fewest were related to health traits (7.3%) (Fig. 3C). Interestingly, further analysis revealed a substantial enrichment of genes enriched in specific GO pathways within the identified QTL regions. Some prominent examples of these enriched genes include *CDH12* (involved in cell adhesion), *CLSTN2* (associated with cell junction organization and synaptic membrane), *EXOC4* (plays a role in synaptic signaling and cell-cell

signaling), and *GRID2* (involved in glutamate signaling). This overlap suggests a potential link between NRNS insertions, gene function in key biological processes, and phenotypic variation.

### Annotating repeats in NRNS

A total of 39.6% of repeats were annotated in the NRNS dataset (Supplementary Table 6). Of the total NRNS, 23.5% were long interspersed nuclear elements (LINEs) and 9.7% were short interspersed nuclear elements (SINEs). The majority of the NRNS dataset was composed of non-repeats (60.4%). When compared with the Brahman reference assembly (GCF_003369695.1), which also has around 53.29% non-repeats and similar percentages of LINEs (27.94%) and SINEs (11.75%), it appears that there is no repeat bias in the NRNS identified in this study.

### Evolutionary analysis

The origin of NRNS was investigated by identifying identical or nearly identical sequences in closely related genomes of the Bos sister species, such as *Bos taurus* (exotic cattle), *Bos gaurus* (Gaur), *Bos frontalis* (Gayal), *Bos grunniens* (Domestic Yak), and *Bos mutus* (wild Yak). A total of 5283 NRNS (40.44%) were identified across these Bos sister species, with each species harboring between 15% and 30% of the total NRNS (Fig. 4A; Supplementary Fig. 4). Interestingly, *Bos taurus*, the sister subspecies of *Bos indicus*, shared the highest number of NRNS not found in other lineages (323) while also having a comparatively lower total number of NRNS compared to its wild and domesticated relatives within the Bos genus. The high number of shared NRNS suggests a closeness of indicus with taurus, but the lower number of total NRNS was unexpected. Additionally, a significant portion of the NRNS was identified only in wild and domesticated yak, suggesting lineage-specific evolution.

**Fig. 5 | Impact of pangenome as a reference for resequencing data analysis. A** Comparison of read mapping percentage for resequencing data (*n* = 30) using a pangenome versus the Brahman reference genome. **B** Number of identified SNPs in 30 cattle samples using pangenome and Brahman reference-based mapping.

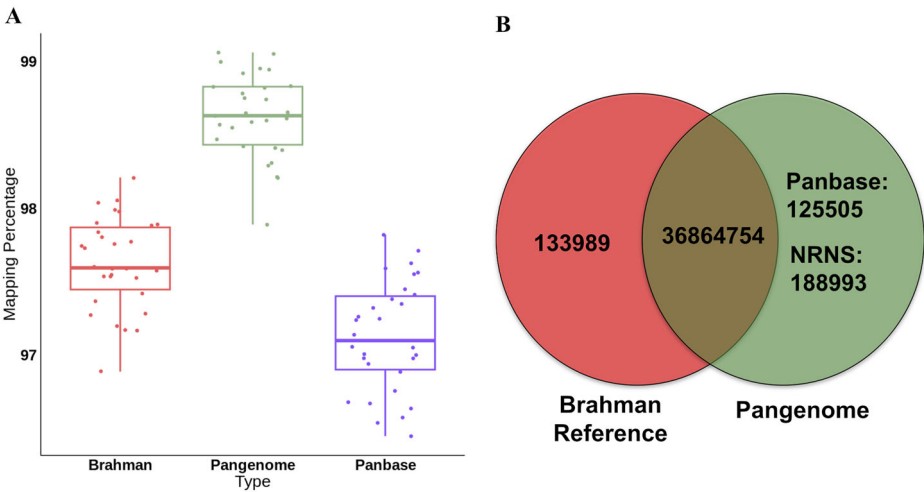

## Discussion

We employed short-read sequencing to identify non-reference sequences circulating within *desi* cattle. Although comparing de novo assemblies from diverse breeds would have been ideal[11,19], generating high-quality assemblies from long reads for numerous Indian cattle genomes is currently limited by cost and resources. Therefore, we opted for an alternative strategy of identifying NRNS from short reads, a method successfully applied in various species like humans, sheep, and European cattle. Our strategy closely resembles the approach used to identify NRNS contigs in the African pangenome, where short reads are mapped to a reference genome, and unmapped reads are assembled into non-reference contigs[12]. To streamline this process, we developed a Bash script named the PanGA pipeline. This pipeline offers a valuable tool for efficiently retrieving NRNS sequences in any diploid species where short-read genome resequencing data exists, but de novo assemblies from multiple individuals are unavailable.

Our PanGA pipeline successfully retrieved 40.9 Mb of NRNS sequences from 68 individuals across 7 diverse *desi* cattle breeds. While this represents roughly 1.5% of the cattle genome, the amount of NRNS identified could have been higher with de novo assemblies. Notably, Dai et al.[19] discovered 125 Mbp of NRNS in their study using 10 de novo assemblies from 10 different Chinese indicine cattle breeds. Similarly, the European cattle pangenome effort identified 85 Mb of NRNS, encompassing 858 individuals from 50 diverse breeds, primarily exotic cattle. The lower amount of NRNS identified in our study compared to others may be attributed to two factors: (1) short-read-based strategies primarily capture longer (>1 kb) NRNS and simpler sequences, and (2) our study included only seven breeds, which is fewer than both the aforementioned pangenome studies. Although NRNS sequences are often enriched for repetitive elements, the sequences identified in this study did not exhibit such enrichment. Reads that map to repetitive regions or paralogous sequences, which contribute to genomic variability, are often part of the pangenome but may map to the same location, making them difficult to distinguish from other mapped reads.

We further explored the identified NRNS sequences to validate their existence and assess their variability within the *desi* cattle population. Interestingly, comparison with existing cattle pangenomes, including one focused on Chinese indicine cattle, revealed limited similarity. This suggests a high degree of *desi* cattle population-specific variation within the large NRNS sequences identified in our study. To further validate the existence of these NRNS sequences within the *desi* cattle genome, we mapped them to available two haplotype resolved de novo genome assemblies of Indian cattle (Tharparkar and Sahiwal) and recently developed assembly of Hainan breed[32], which represents Chinese indicine cattle. We found a substantially higher number of matches. This confirms the presence of these NRNS

Further investigation extended to searching for NRNS sequences within the genomes of other Bovidae family members, including *Bison bison* (American bison), *Bubalus bubalis* (water buffalo), *Tragelaphus oryx* (eland), *Ovis aries* (sheep), and *Capra hircus* (goat). Only two NRNS were found across all these species. Interestingly, the most shared NRNS were with *Bison bison*, followed by buffalo, eland, sheep, and goat (Fig. 4B, Supplementary Fig. 5). Similar trends were observed in having unique NRNS (shared only between *desi* cattle and the species) in these species. This trend also correlates well with the inferred evolutionary divergence times based on phylogenetic analysis, with the highest number of shared NRNS occurring within taxonomically closer species.

### Assessing the utility of a pangenome as a reference

We compared the mapping efficiency of short reads to a Brahman reference genome (GCF_003369695.1) and a pangenome reference. The pangenome was constructed by integrating the base assembly (the Brahman assembly, referred to as panbase) with 13,065 NRNS (~41 Mbp).

The pangenome consistently yielded a higher mapping ratio across all 30 samples in the study (Supplementary Data 5). The average mapping rate increased from 97.29% using the Brahman reference to 98.62% using the pangenome (Fig. 5A). While the panbase within the pangenome also demonstrated high mapping efficiency (97.13%), a slight decrease compared to the standalone Brahman reference was observed. This may be attributed to the presence of NRNS which have improved the overall alignment accuracy, resulting in fewer reads mapping to the panbase due to their correct placement on NRNS. We also observed an average of 1% increase in properly paired reads across the datasets using the pangenome compared to the Brahman reference. However, analysis of aligned reads revealed comparable read mapping quality between the pangenome, panbase, or the Brahman reference.

Furthermore, SNP analysis using the Brahman reference genome identified a large number of spurious SNPs (133,989). These spurious SNPs were absent when using the pangenome for SNP calling. Additionally, the pangenome analysis revealed 314,498 novel SNPs not identified with the Brahman reference (Fig. 5B). These novel SNPs likely represent true genetic variation within the *desi* cattle population. Notably, panbase identified 125,505 of these novel SNPs, while the NRNS contigs within the pangenome harbored an additional 188,993 novel SNPs. Out 188,993 novel SNPs, 35,099 SNPs were identified in placed contigs (1850 contigs), while 153,894 SNPs were found in unplaced contigs. This breakdown highlights the contribution of both panbase's alignment capabilities and the inclusion of NRNS sequences in the pangenome for comprehensive SNP discovery.

sequences and highlights their potential significance for *desi* cattle biology. Additionally, analysis of NRNS presence across the 68 individuals revealed that a large proportion (35.7%) of NRNS as private insertions, existed in only one individual. This finding aligns with observations from the African pangenome study[12]. Another limitation of short read based strategy is to place the NRNS onto chromosomes. Like other short read based studies, only few percent of NRNS were placed confidently onto reference sequences.

From previous studies, it is well known that NRNS are important as they can have coding potential[11,12,16]. In this study, we also identified coding sequences in NRNS using ab initio and evidence-based methods utilizing RNA-seq data. Mapping RNA-seq data to NRNS suggests potential coding ability and provides evidence for their expression in the *desi* cattle population. Additionally, research by Leonard et. al.[31] supports the notion of differential expression for these genes. Further, the presence of NRNS within genic regions strengthens the hypothesis of variable genic sequences within the population[12].

Functional annotation using Gene Ontology (GO) analysis of proteins potentially associated with the identified NRNS sequences revealed enrichment in terms associated with critical biological processes. Notably, several genes enriched in the GO analysis were also found to reside within major QTL regions, and previous studies have demonstrated their roles in affecting traits. For example, the *CDH12* gene, enriched for cell adhesion related GO terms, resides within QTL regions associated with milk composition traits in Holstein cattle as well as subcutaneous fat thickness QTL[33]. Similarly, *CLSTN2*, enriched for similar GO terms and associated with the synaptic membrane, is linked to udder attachment and calving ease QTLs. Studies in goats and sheep suggest its potential role in litter size[34,35]. *EXOC4*, enriched for synaptic and cell-cell signaling terms, is associated with reproductive traits such as age at first calving in Canchim beef cattle[36]. *GRID2*, enriched for synaptic signaling and glutamate receptor activity, reported for production and caracas trait in beef cattle[37] and also involved in the sexual maturity of Simmental cattle[38]. These examples highlight the potential of NRNS to influence economically important traits in *desi* cattle. Functional validation of these enriched genes and their association with NRNS insertions will be crucial for elucidating the precise mechanisms by which NRNS contributes to phenotypic diversity.

Our analysis of NRNS origin revealed a substantial proportion (40.44%) shared across Bos sister species, suggesting an ancestral origin for a large portion of the *desi* cattle pangenome. This aligns with observations in human pangenome studies, where a majority of non-reference sequences were identified as ancestral sequences[11]. Interestingly, while *desi* cattle shared the most unique NRNS sequences with *Bos taurus*, as expected for closely related species, *Bos taurus* exhibited a surprisingly lower total number of NRNS. This observation suggests a potential loss of ancestral NRNS sequences within the *Bos taurus* lineage, likely due to artificial selection pressures during domestication[39]. Alternatively, Dai et al.[19], has showed that about 10% of NRNS in *Bos indicus* were due to introgression from other species. These introgressions would have occurred after the divergence of the *Bos taurus* and *Bos indicus* lineages. Furthermore, the limited number of NRNS shared across Bovidae family members, with a decreasing trend with increasing evolutionary divergence, reinforces the link between taxonomic closeness and shared NRNS sequences[11]. These findings provide valuable insights into the unique genetic makeup of *desi cattle* and suggest lineage specific selection of NRNS in shaping the genome.

Consistent with findings in human[40], goat[14], pig[16], and other cattle studies[18,41], our results demonstrate the superiority of pangenome references for *desi* cattle genomics. The inclusion of NRNS sequences in the pangenome reference facilitates improved mapping of short reads by enabling alignment to previously unmapped regions, potentially present within NRNS.

Interestingly, we observed a decrease in the mapping ratio of panbase compared to the standalone Brahman reference. This decrease may be attributed to the presence of NRNS, which have improved overall alignment accuracy. As a result, fewer reads map to panbase due to their correct placement on NRNS. This enhanced mapping accuracy translates into a reduction in false alignments, a reduction of spurious SNPs, and an increase in true SNPs within panbase[16,18]. Furthermore, the availability of NRNS allows for the identification of novel SNPs that would otherwise remain undetected with a reference genome alone. Since SNPs are crucial markers for various genomic studies (such as genome-wide association studies (GWAS), marker assisted selection (MAS), and population diversity analysis), capturing the full spectrum of NRNS is essential for a comprehensive understanding of *desi* cattle genetic variation.

This enhanced mapping efficiency translated into a reduction in false alignments and spurious SNPs within panbase, the Brahman reference-based portion of the pangenome.

## Conclusion

This study comprehensively investigated the genomic landscape of *desi* cattle by identifying and characterizing NRNS. The development of the PanGA pipeline facilitated the efficient detection of NRNS, revealing a substantial proportion within the *desi* cattle genome. Our analysis revealed a high degree of variability in NRNS across the *desi* cattle population, suggesting their potential role in shaping phenotypic differences. Furthermore, evidence of coding potential within NRNS, as well as their presence in both coding and non-coding regions, underscores their diverse functional significance. The ancestral nature of a large proportion of NRNS, as evidenced by their presence in related Bos species, highlights the importance of pangenome approaches in capturing the full spectrum of genetic variation. Finally, the pangenome's ability to improve read mapping, reduce spurious SNPs, and identify novel SNPs underscores its superiority over traditional reference genomes for comprehensive genomic studies.

## Methods
### Ethics statement
Samples were obtained in strict accordance with the guidelines set forth by the Committee for the Purpose of Control and Supervision on Experiments on Animals (CPCSEA), India, and with the approval of the Institutional Animal Ethics Committee (IAEC) of the National Institute of Animal Biotechnology. We have complied with all relevant ethical regulations for animal use. Blood samples were collected from cattle older than one year via jugular venipuncture, a standard and minimally invasive procedure for blood collection in large animals.

### Sample collection and sequencing
To construct a pangenome of *desi* cattle, whole-genome sequencing was conducted on individuals from seven different breeds. The sample set included 16 individuals from the Gir breed, 15 from Sahiwal, 14 from Tharparkar, 9 from Hariana, 6 from Red Sindhi, 6 from Kankrej, and 2 from Ladakhi breeds, with all individuals being female. Genomic DNA extraction was performed from the blood samples using the QIAamp DNA Mini Kit (Qiagen). The quality and quantity of extracted DNA were evaluated using a NanoDrop spectrophotometer. High-quality DNA samples were subsequently sent to Agrigenome and Neuberg Supratech for sequencing on the Illumina platform. Genomic DNA libraries was prepared for sequencing using the KAPA HyperPlus Kit (Roche, #07962428001). Libraries were barcoded with unique dual-index (UDI) adapters from IDT for Illumina TruSeq. The libraries were quantified using a Qubit 4.0 Fluorometer (Thermo Fisher Scientific, #Q33238) with the Qubit dsDNA HS Assay Kit (Thermo Fisher Scientific, #Q32851). Quality assessment was performed using the TapeStation 4150 (Agilent). Sequencing was conducted on an Illumina NovaSeq 6000 platform using paired-end 150 bp chemistry. This generated paired-end (PE) sequencing data with a read length of 150 bp for each sample.

### Development of the PanGenome Analysis (PanGA) pipeline
Identifying NRNS from short reads is a multi-step process. This study adopts a framework similar to that used by Sherman et al.[12] for identifying

novel contigs in the African human pangenome. To streamline the process and enable its application to other genomes, we developed the PanGA pipeline (https://github.com/GCBL-NIAB/PanGA_Pipeline). PanGA is written in Bash scripting and integrates essential bioinformatics tools, including fastp v0.23.3 (RRID:SCR_016962)[42], Bowtie2 v2.3.5.1 (RRID:SCR_016368)[43], Samtools v1.2 (RRID:SCR_002105)[44], MaSuRCA v4.0.3 (RRID:SCR_010691)[45] and CD-HIT v4.8.1 (RRID:SCR_007105)[46] (Fig. 1). The pipeline consists of the following steps:

- *Preprocessing:* This initial step focuses on quality control, aimed at discarding poor-quality reads from the raw FASTQ data. Since Illumina paired end data with a constant read length was utilized in the study, fastp was employed with optimized parameters to preprocess each sample efficiently using multiple threads. During this step, each read is processed for adapter sequences, undergoes quality filtering and trimming. Additionally, reads below the length threshold (50 bp) are removed. The output of this preprocessing is a set of high-quality reads suitable for subsequent pipeline steps.
- *Alignment:* All the high quality reads obtained in the previous step were aligned onto the reference genome. In this study we have used Brahman genome assembly (GCF_003369695.1) as a reference genome for *desi* cattle. The tool used for alignment in this study was Bowtie2. The alignment outputs were generated in SAM format and subsequently converted to compressed BAM format.
- *Extraction of unaligned reads:* The alignment file (SAM/BAM) contains both reads mapped to the reference genome and unaligned reads. Samtools was employed to extract different categories of unaligned reads from BAM file based on their mate's mapping status:

Completely unaligned read pairs (both mates unmapped) were extracted using samtools with option "fastq -f 12"

Forward reads (R1) where the mate was mapped were extracted using samtools with option "fastq -f 68 -F 8"

Reverse reads (R2) where the mate was mapped were extracted using samtools with option "fastq -f 132 -F 8"

These flags ensured only unmapped reads were extracted while considering their mate's mapping status. All extracted unaligned reads were saved in FASTQ format.

- De novo *assembly of unaligned reads:* The unaligned reads from each sample were assembled using the Masurca assembler. Masurca requires a configuration file where data details and various parameters are specified. The assembled contigs from Masurca were then filtered based on a minimum length threshold of 1000 bp. These de novo assembled contigs were considered novel contigs specific to each sample.
- *Contamination filtering:* The de novo assembled contigs underwent further analysis to identify potential contamination by matching them with the non-redundant nucleotide database of NCBI. This process involves several steps, including aligning the contigs with the database and identifying their lineage. To streamline this process, we developed a new pipeline called Fast2Lineage, which integrates these steps. Using Fasta2Lineage (https://github.com/GCBL-NIAB/Fasta2Lineage), we assigned a lineage to each contig of the sample. Contigs that did not show any alignment were considered novel, while those aligning with sequences from Chordata were classified as cattle contigs. However, any contigs showing alignment with sequences from archaea, bacteria, fungi, plants, or any other phylum outside of Chordata were flagged as potential contamination and subsequently discarded.
- *Generation of Non-Redundant NRNS:* To generate a non-redundant set of NRNS, contigs from each sample that passed contamination filtering were clustered using CD-HIT[46]. This step effectively removes sequence redundancy within the putative NRNS set. We utilized the cd-hit-est command with the following parameters: -aS 0.8, -g 1, -c 0.9, and -M 0. CD-HIT produces a non-redundant set of contigs considered as NRNS existing in the *desi* cattle population. These NRNSs serve as the final output of the PanGA pipeline and can be employed alongside the reference genome to construct the pangenome.

## Development of Fasta2Lineage tool

The Fasta2Lineage pipeline represents a comprehensive tool designed to annotate the lineage of input fasta sequences by integrating the NCBI non-redundant nucleotide (nt) database, NCBI taxonomy database, GenBank accessions data using BLAST v2.9.0+ (RRID:SCR_004870)[47], Perl and bash scripts (Supplementary Fig. 5). The installation prerequisites include ncbitax2lin[48], which dumps taxonomy files into comma-separated lists of lineages for each taxon ID. Additionally, a list of Accession and corresponding taxon ID is required for transferring taxon IDs of each sequence in the nt database. BLAST is essential for matching input sequences to the nt database. The pipeline accepts input sequences in fasta format, performs BLAST with the nt database, selects the accession IDs of matched sequences, and subsequently identified taxa IDs. It then reports the lineage of each sequence by extracting the lineage of each taxon ID, providing comprehensive annotation for fasta sequences.

## Comparison with published pangenomes and other genome assemblies

We downloaded two published pangenomes of cattle from Zhou et al.[20] and Dai et al.[19]. NRNS sequences were compared against these pan-genomes using strict criteria, requiring a minimum of 95% coverage and 95% identity, utilizing BLASTn. The results indicated the presence of NRNS in these other cattle pan-genomes. Furthermore, two haplotype-resolved genome assemblies of Tharparkar (GCA_029378745.1) and Sahiwal (GCA_029378735.1) breeds were downloaded from NCBI. We also downloaded a genetically distant *Bos indicus* assembly of the Hainan breed (GCA_039881165.1), which represents Chinese indicine cattle[32]. NRNS sequences were searched against these genomes using BLASTn with the same parameters of 95% coverage and 95% identity.

## Calling presence/absence of NRNS per sample

To identify NRNS present in each sample, de novo assembled contigs were aligned against the reference NRNS set using BLASTn. The BLAST output was filtered requiring >90% sequence identity and >80% sequence coverage thresholds. Contigs that met these criteria were considered present NRNS in the corresponding sample and assigned a "1" in the genotyping matrix.

## Placement of NRNS onto the reference genome

We utilized Popins2 v0.13.0[49] to map NRNS onto the reference genome. This process involved employing submodules of Popins: "assemble," "contigmap," and "place" to determine the locations of NRNS. Initially, Popins was run on the BAM files generated by the PanGA pipeline. The "assemble" module was used to extract unaligned and poorly mapped reads for each sample. Subsequently, the "contigmap" submodule mapped these reads onto the NRNS. Mate information was merged and saved as "location.txt". Next, the "place" submodule was used to anchor the NRNS to the samples. Finally, all locations from successful placement of ends of NRNS onto reference were written to a VCF file.

Further, we screened the VCF file using stringent criteria to classify the placement of NRNS into three categories: one end placed, two ends placed, and unplaced. The criteria were as follows:

1. The placement of ends of NRNS were considered valid only if the anchoring read pair (AR) count is greater than one.
2. The anchorpoint reported in the VCF file should be within 500 bases of the end of a NRNS.
3. Any end of NRNS placed more than once on the reference genome is regarded as unplaced.
4. If both ends of NRNS are aligned onto the reference genome, they should be in the same direction and on the same chromosome.

Finally, all ends of NRNS are chromosome unambiguous and region unambiguous, otherwise considered as unplaced.

To further investigate the factors influencing NRNS placement, we analyzed the presence of Tandem Repeats (TRs) within NRNS sequences.

TRs were identified within NRNS using Tandem Repeat Finder (TRF) with a score threshold of >100.

## Transcription potential of NRNS

We employed both ab initio and evidence-based methods to assess the transcriptional potential of NRNS. For ab initio prediction, novel genes within NRNS were identified using Augustus v3.4.0 (RRID:SCR_008417)[50] with the options '--singlestrand=true --genemodel=complete'. In parallel, an evidence-based approach was employed by mapping RNA-Seq data from 47 samples from 47 individuals representing various *desi* cattle breeds (such as Gir, Tharparkar, Sahiwal, and Hariana) onto the Brahman reference genome (GCF_003369695.1). The STAR aligner v2.7.10 (RRID:SCR_004463)[51] was used to map RNA-Seq reads onto the reference sequences, one sample at a time. Subsequently, unmapped reads from all 47 samples were merged into single read1 and read2 files. These merged reads were then realigned to the Brahman reference genome without supplying the GTF file. Following this realignment, the merged unmapped reads were aligned to the NRNS using STAR. Finally, Stringtie v2.1.1 (RRID:SCR_016323)[52] was applied to the resulting BAM file to identify potential transcripts for each gene locus. To identify full-length genes and annotate their function, peptides predicted from Augustus were first mapped to the Bovidae family proteome using BLASTp. Alignments were filtered using the parameters of sequence identity >= 80%, sequence coverage >= 50%, and e-value of <= 1e-6. Peptides that could not be aligned were further mapped to the chordata subset of the nr database and filtered using the same parameters. Subsequently, the same process was repeated for identified transcripts from Stringtie using BLASTx. This comprehensive approach allows for the functional annotation of predicted genes within NRNS, providing insights into their potential roles and impacts.

## GO enrichment analysis

A non-redundant set of genes associated with NRNS identified in the study was subjected to functional annotation and enrichment analysis using the clusterProfiler package (RRID:SCR_016884)[53] from the Bioconductor in R (RRID:SCR_006442). A significance threshold of p-value <= 0.05 (FDR by Benjamini–Hochberg) was applied to identify significantly enriched terms with reference to the GO annotation database of *Bos indicus* (https://github.com/ASBioinfo/Genome_wide_annotation_Bos_indicus). The top 10 most significantly enriched terms were then visualized for each GO category: biological process, molecular function, and cellular component.

We also performed GO term analysis for genes associated with NRNS contigs that were found in only a few samples (cloud-like NRNS) versus those present in most samples (core NRNS). Core NRNS contigs were defined as those present in approximately 95% or more of the samples (i.e., in ≥64 out of 68 samples), while cloud-like NRNS contigs were defined as those present in ≤5 out of 68 samples. Genes from these contigs were analyzed for GO enrichment using the method described above.

## QTL analysis

To identify potential NRNS influencing genes located within major QTLs of cattle, we downloaded the annotation of the ARS_UCD 1.2 genome assembly[54]. Subsequently, the nonredundant set of genes associated with NRNS underwent a BLASTp search against proteins annotated in the ARS_UCD 1.2 genome assembly. The identified orthologous genes were then cross-referenced against a publicly accessible cattle QTL database available at https://www.animalgenome.org/. This analysis allowed us to identify enriched QTLs and their associated traits.

## Annotating repeats in NRNS

The composition of repeats and transposable elements (TE) within the NRNS was analyzed using RepeatMasker. The tool was executed with the parameters '--species cow', '-xsmall', and '-nolow'. RepeatMasker v4.1.0 (RRID:SCR_012954; http://www.repeatmasker.org) generated a summary table providing a detailed breakdown of the various repeat types and their abundance within the NRNS set.

## Evolutionary analysis

To establish the evolutionary connection between the identified NRNS in our study and species of the Bos genus, we downloaded the reference genomes of *Bos frontalis* (GCA_007844835.1)[55], *Bos gaurus* (GCA_014182915.2)[56], *Bos grunniens* (GCA_027580245.1)[57], *Bos mutus* (GCA_027580195.1)[57], and *Bos taurus* (GCF_002263795.2)[54]. Additionally, we explored the presence of NRNS in other members of the Bovidae family, including *Ovis aries* (GCF_016772045.1)[58], *Capra hircus* (GCF_001704415.2)[59], *Bison bison* (GCA_000754665.1)[60], *Tragelaphus oryx* (GCA_006416875.1)[61], and *Bubalus bubalis* (GCF_003121395.1)[62]. NRNS sequences were aligned against these reference genomes, and alignments meeting or exceeding 95% identity and 95% query coverage were regarded as true hits. These hits suggest the presence of similar sequences in related species, potentially indicating conserved regions or a shared evolutionary history.

## Assessing the utility of a pangenome as a reference

To evaluate the suitability of the pangenome as a reference for *desi* cattle genetic studies compared to a standard single-reference genome, we adopted a comprehensive approach. The pangenome was constructed by integrating the base assembly (the Brahman assembly, referred to as pan-base) with 13,065 NRNS (~41 Mbp). Initially, short-read Illumina sequencing data from 30 *desi* cattle were aligned to both the Brahman reference genome (GCF_003369695.1) and our developed pangenome. Subsequently, we assessed the mapping quality and mapping rate (percentage of successfully mapped reads) for each sample across both references. Next, to assess the impact of the pangenome on SNP calling, we utilized the alignment files generated from mapping each sample to both the pangenome and the Brahman reference genome. Duplicate reads were removed using Picard Tools v3.1.1 (RRID:SCR_006525; https://broadinstitute.github.io/picard/), and SNP calling was executed using GATK v4.3.0.0 (RRID:SCR_001876; https://gatk.broadinstitute.org/hc/en-us). We applied rigorous criteria to filter the identified SNPs, including average variant quality score (QUAL) > 30, variant confidence/quality by depth (QD) > 2, and RMS mapping quality (MQ) > 20. Lastly, we compared the resultant SNP calls obtained from both the pangenome and the Brahman reference to determine if the pangenome offered a more suitable reference for *desi* cattle genetic studies.

## Statistics and reproducibility

Statistical analyses were performed using the cited software packages. Complete details regarding the Illumina short-read sequencing data are accessible in the data availability section. To ensure consistency and replicability, we utilized the software tools and codes mentioned in the methods and code availability sections for all analyses. All parameters used in our analyses and data processing are comprehensively documented in the methods section.

## Reporting summary

Further information on research design is available in the Nature Portfolio Reporting Summary linked to this article.

## Data availability

The short-read sequencing dataset generated from the Illumina platform and used in this study has been submitted to the Indian Biological Data Centre (IBDC). The accession numbers for these datasets are detailed in Supplementary Table 7. Similarly, the accession numbers for the RNA-seq data used in the study are provided in Supplementary Table 8. All the data can also be accessed from the NCBI.

## Code availability

The tools developed and used in this study for constructing the pangenome (PanGA Pipeline) and for contamination filtering (Fasta2Lineage) are available on GitHub at the following repositories: PanGA Pipeline: https://github.com/GCBL-NIAB/PanGA_Pipeline, Fasta2Lineage: https://github.com/GCBL-NIAB/Fasta2Lineage

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

## Acknowledgements

This study was supported by the Department of Biotechnology (DBT), India, under two projects: Identification of Key Molecular Factors Involved in Resistance/Susceptibility to Paratuberculosis Infection in Indigenous Breeds of Cows [BT/PR32758/AAQ/1/760/2019], and Genomics for Conservation of Indigenous Cattle Breeds and for Enhancing Milk Yield, Phase-I [BT/PR26466/AAQ/1/704/2017]. While the transcriptomics data were specifically produced under the project "Identification of key molecular factors involved in resistance/susceptibility to paratuberculosis infection in indigenous breeds of cows" [BT/PR32758/AAQ/1/760/2019]. The authors gratefully acknowledge the financial support from DBT, Ministry of Science,

New Delhi, India. We also extend sincere thanks to the National Institute of Animal Biotechnology (NIAB) for their invaluable support throughout this study. In particular, S.A. expresses deep gratitude to Dr. G. Taru Sharma, Director of NIAB, for her support. M.N., C.P.V.T., and B.D.R. were supported by the appropriated project 8042-31000-112-000-D, "Accelerating Genetic Improvement of Ruminants Through Enhanced Genome Assembly, Annotation, and Selection" of the USDA Agricultural Research Service. Any mention of trade names or commercial products is solely for the purpose of providing specific information and does not imply recommendation or endorsement by the U.S. Department of Agriculture. The USDA is an equal opportunity provider and employer.

## Author contributions

S.A., B.D.R., S.N.R., and S.S.M. designed the study. S.S.M., R.K.G., and S.A. facilitated Sample collections and sequencing. S.A and N.K.P. assembled the genomic sequences and established the contamination screening and PanGA pipeline. S.A., N.K.P., and A.S. analyzed the data for NRNS identification through PanGA pipeline, Genotyping of NRNS in the population, TE analysis, GO analysis and evolutionary analysis. M.N. performed QTL analysis. R.K.G., C.P.V.T., B.D.R., and S.A. drafted the manuscript. S.S.M., S.N.R., and C.P.V.T. edited the manuscript. All authors reviewed the final manuscript before submission.

## Competing interests

The authors declare no competing interests.
