## [Transparent Peer Review file · Communications Biology]

Constructing a Draft Indian Cattle Pangenome Using Short-Read Sequencing

Corresponding Author: Dr Benjamin Rosen

Version 0:

Reviewer comments:

Reviewer #1

(Remarks to the Author)

This manuscript describes the characterization of non-reference novel sequences (NRNS) discovered by aligning short read sequences from 68 cattle representing 7 desi breeds to the long-read Brahman assembly. There was little overlap of NRNS identified in the recent Asian cattle pangenome, highlighting the importance of investigations like this to capture more of the genetic diversity of worldwide cattle breeds.

1. Line 119: Although it's certainly not a main point of this paper, 20% contamination is substantial and because your script classified the source contaminants, it would be useful to add a sentence reporting the main type(s) of contamination observed. See, for example, Whitacre et al. 2015. BMC Genomics 16:1114.
2. Line 153: Only 17.61% of NRNS were mapped to chromosomes. Did the rest map to multiple locations (per #3 of the VCF filtering) or did some not map at all, and if so what was the proportion of each? If these NRNS are mapping to multiple locations, do they match interspersed repeats? Is this low percentage of placing them on chromosomes a function of being based on short reads? Were similar low percentages seen in other studies using short reads?
3. Line 174-175: Are the matches to a particular protein domain rather than specific proteins?
4. Line 229-234 and 334-342: The Asian pangenome paper (Dai et al., 2023) showed that about 10% of NRNS were due to introgression from other species. To explain the NRNS finding for *Bos taurus*, rather than acquiring the sequences and then losing them through artificial selection, isn't a more parsimonious (or at least another) possibility that during their evolution the European *Bos taurus* cattle were never exposed to individuals from those *Bos* species and admixture due to inter-species introgression (for the highlighted species) occurred only in the *Bos indicus* lineage?
5. Line 262: How many of the 188,993 novel SNPs were in placed vs. unplaced contigs? Are the ones in unplaced contigs also potentially spurious, especially if they are unplaced because they map to multiple regions of the genome?
6. Line 472: Why is a more relaxed sequence identity threshold being used within breed (90%) than between breeds (95%)? This seems backwards.
7. Line 579: Where is the VCF file deposited?
8. Figure 3: What does black and gray on the ideograms represent? Are the chromosomes oriented with centromere to the left? About half of the Brahman reference sequences are backwards compared to ARS-UCD1.2, which are oriented with centromeric end as 0 Mb. Has this been taken into account?
9. Supplementary Tables: Recommend sorting all tables by breed.
10. Supplementary Table 3: The "Seq after Lineage" and "Total Seq" columns are identical. Should they be? If so, delete one.
11. Supplementary Table 4: The layout of this table is confusing. Suggest splitting or reformatting. What does "Not hit" mean?

Minor

12. Line 57: Use *Bos taurus indicus* throughout or add *Bos indicus* to the known as clause here.
13. Line 89: What does this mean: "mapping additional population genomes." Recommend rewording for clarity.
14. Line 110: In Supplementary Table 2 the number of PE reads is 21,893 million. Delete "2 x"
15. Line 111 and throughout: Insert the version of the Brahman reference used. Based on the code for the PanPA pipeline it was UOA_Brahman_1 (GCF_003369695.1).
16. Line 114: Reword to clarify that the de novo assemblies were performed only for the unaligned reads.
17. Line 116: Reword sentence. I initially interpreted "across all samples" to mean you found the same 1000 bp contigs in

every sample.

18. Line 171 and Supplementary Figure 2: StringTie is not represented in the figure.

19. Lines 205, 298, 301: Is significant being used in a statistical sense. If so, report a P-value.

20. Line 380: Were all samples female?

21. Line 384-386: What library preparation methods were used? Barcoding? Same chemistry for both INRP000053 and INRP000159?

22. Line 443: Add a citation for CD-HIT.

23. Line 454: Add a citation for ncbtax2lin

24. Line 550: *Bos taurus* is a subspecies not sister species.

25. Table 1: Add a footnote listing the accession numbers for the Tharparkar and Sahiwal assemblies.

Reviewer #2

(Remarks to the Author)

Azam et al. present a compelling manuscript analysing nonreference insertions across a large set of diverse Indian cattle breeds, with a detailed analysis on possible functional consequences of such sequence. The results and conclusions are especially important to those in the cattle field but are of general interest to all species where generating long read assemblies for graph pangenomes is as feasible. I find the manuscript overall extremely convincing, although I have several suggestions to address.

Major comments:

After a quick analysis of the NRNS with TRF, almost 42% of the sequences contain tandem repeats (including 2823 out of 13065 with scores >100). These sequences are especially challenging to map, so I think it would be worth analysing how many of these tandem-repeat containing NRNS were amongst the unplaceable NRNS. Interestingly, in the Zhou et al. VCF, only 2259 out of 22324 were TRs with scores >100, so your set of NRNS seems to be capturing much more tandem-repeat variation than previous short read SV pangenomes. Maybe this is a strength of PanGA compared to earlier approaches and could be emphasised?

Removing contaminants based on databases may be a useful step if you are uncertain about the purity of the sample, but nonreference sequence frequently is identified as “contaminants” due to recurring reference-bias (see Rhie et al. <https://www.nature.com/articles/s41586-023-06457-y> in the section: “Contamination of genomic databases”). I find nearly 20% of your NRNS showing up as “contaminants” as high, but may seem reasonable to you based on the quality of your sample preparation?

Given the large improvement in overlapping NRNS against the Tharparkar and Sahiwal assemblies compared to the short read SV VCFs, I would be interested in seeing a repeat of that analysis against e.g., the Hainan indicine assembly from Xia et al. (<https://link.springer.com/article/10.1186/s13059-023-03052-2>). This would help resolve if it is divergence/evolution related (if a low overlap of NRNS with the Hainan assembly) or VCF vs assembly related (if the overlap level is closer to the Tharparkar/Sahiwal assemblies). Especially given Tharparkar and Sahiwal make up around 40% of your samples, I think comparing to an evolutionary-similar but not identical breed is important for this analysis.

I think some of the “pangenome” methods would benefit from slightly clearer descriptions when first mentioned. I was unclear what “PanBase” referred to while reading the results and only understood after reading the methods later on. Also when you say “augmenting” the reference, does this mean you concatenated the files together (effectively more unplaced contigs), or is this a graph-style augmentation (like done by vg augment)? Some clarification here would be useful. I’m also somewhat uncertain on how augmenting the linear reference with NRNS would lead to novel variant calls on the sequence that was always present in the linear reference, but maybe a decrease in mismapped reads would improve GATK’s ability to call variants.

Minor comments:

There are many spacing issues (double spaces between words, no spacing after periods, inconsistent spacing before references, etc), that could be fixed.

- Remove “staggering” and “impressive” adjectives
- It is not obvious until the discussion that PanGA was created for this manuscript, so maybe clarify that you use “our PanGA pipeline” rather than “the PanGA pipeline”.
- It may be coincidence, but Ladakhi are the smallest sample size (n=2) and Gir are the largest (n=16), so is observing the fewest/most breed-specific NRNS respectively just a consequence of sample size? Red Sindhi seems to be the main outlier to this, as it has a lot of breed-specific NRNS but only n=6.
- GO enrichment is easy to over-interpret, but I would be curious to see if there is a difference in GO terms for NRNS-genes found in only a few samples versus many (i.e. cloud- vs core-like genes)
- Clarify if “23.5% were long interspersed nuclear elements” refers to 23.5% of 39.6% (i.e. 9.3% of the total NRNS) or if it is 23.5% of the total NRNS.
- “Identifying identical sequences” the methods describe 95% similarity as the threshold, not identical.
- “Exotic cattle” seems to refer to *bos taurus taurus*, is this correct? I’m not familiar with that terminology, but I was expecting

this to be something uncommon rather than the cattle reference genome.

- “Additionally, research by 31 supports...” Reference 31 has no RNA-seq component, so maybe this was referring to a different Leonard et al (<https://link.springer.com/article/10.1186/s13059-023-02969-y>) that did analyse RNA-seq and non-reference sequence?
- The GO term discussion paragraph is interesting, but ultimately quite long and purely speculative based on references to other papers. Perhaps it can be shortened or only keep a few examples to keep the focus on what the novel results in this manuscript are.
- It is likely beyond the scope of what should be done, but I wonder if the NRNS primarily assembled from short reads where one of the read-pairs mapped to the reference had a higher chromosomal placement rate. This would effectively be some type of “flanking sequence”, so would make sense for higher placement rates compared to completely unmapped reads.
- Figure 1. Drop the n=68 as that is dependent on the input, not the pipeline. Add n=68 to left part and ideally the n= for each breed.
- Fig 2. Consider if this should be a Sup Figure. PanGA is a useful contribution to the field, but the primary output of this paper is not the pipeline itself (to my understanding).
- Figure 4A: what does BP/CC/MF refer to? Please describe in the caption.
- Figure 5. Please create this as an upsetplot similar to Supplementary Figure 1, as a 5-way Venn diagram is not easy to interpret.
- Table 2 and 3. These could be combined into one table. Private-only is only in Table 3, but the only reference to private-only is related to Table 2? Can move the “(of 68)” to the caption rather than listing in every row.
- Sup Fig 2 is not particularly clear compared to Fig 1 where you provide information on which steps are tools/output/filtering. The annotation is already fully described in the methods, so I would suggest cutting this supplementary figure.
- Sup Fig 3 is effectively just a less detailed version of Sup Table 5, which is already referenced in the previous sentence, so I would suggest cutting.

Version 1:

Reviewer comments:

Reviewer #1

(Remarks to the Author)

The authors prepared a careful response to the initial review of their manuscript. The additional analyses undertaken and additions made to the manuscript appropriately address all the major concerns identified. Readability is also improved by the authors' attention to the minor corrections suggested. The results reported are an important contribution to our understanding of genetic diversity of worldwide cattle breeds.

Minor

Supplementary Figure 3. The diamonds in the GO plot all appear to be the same size and are the same color. The figure legend suggests variability in size and color is possible. And is it expected in A that they have the same gene ratio? Please double check that this figure has been generated correctly.

Reviewer #2

(Remarks to the Author)

The revised manuscript is largely improved and all of my comments have been satisfactorily addressed.

Reviewer #1 (Remarks to the Author):

This manuscript describes the characterization of non-reference novel sequences (NRNS) discovered by aligning short read sequences from 68 cattle representing 7 desi breeds to the long-read Brahman assembly. There was little overlap of with NRNS identified in the recent Asian cattle pangenome, highlighting the importance of investigations like this to capture more of the genetic diversity of worldwide cattle breeds.

1. Line 119: Although it's certainly not a main point of this paper, 20% contamination is substantial and because your script classified the source contaminants, it would be useful to add a sentence reporting the main type(s) of contamination observed. See, for example, Whitacre et al. 2015. BMC Genomics 16:1114.

Authors' response: We would like to clarify that a high percentage of reads in each sample were mapped to the reference genome. In fact, the overall alignment percentage to the reference genome ranged from a minimum of 95.86% to a maximum of 99.24% (Line: 115). When averaged across all samples, 98.59% of the data were aligned. We extracted the unaligned reads, which represent an average of 1.41% of the total reads. On average, 20% of these assembled contigs represent approximately 0.3% of the original data. This indicates that only a very small percentage of the data exhibited potential contamination.

As suggested, we have classified the contaminants and details have been provided in Supplementary Table 4. We have also modified the Line 124-127 as follows: "A cumulative total of 50,229 contaminant sequences, primarily consisting of environmental sequences (61.48%), *Theileria annulata* (24.72%), *Theileria orientalis* (7.87%), *Babesia bigemina* (1.22%), and other minor contaminants below 1%, were removed from the sequence data, leaving 194,126 de novo contigs for subsequent processing."

2. Line 153: Only 17.61% of NRNS were mapped to chromosomes. Did the rest map to multiple locations (per #3 of the VCF filtering) or did some not map at all, and if so what was the proportion of each? If these NRNS are mapping to multiple locations, do they match interspersed repeats? Is this low percentage of placing them on chromosomes a function of being based on short reads? Were similar low percentages seen in other studies using short reads?

Authors' response: Out of 13,065 NRNS contigs, 2,302 (17.61%) were successfully placed on chromosomes. The majority of the NRNS contigs, 10,670 (~81.6%), were not assigned to chromosomes. Additionally, a small fraction, 93 contigs (0.7%), were discarded due to multiple mappings or inconsistencies in mapping direction. (Line: 166-170)

The low percentage of NRNS contigs being placed on chromosomes is primarily due to the origins of the short-read sequence data used to assemble the contigs. The sequences were originally unmapped, or only had a single end mapped. Once assembled, it is only those reads with a mapped pair, or partial mapping which could be used to place the contigs. Short reads provide limited context for mapping contigs, particularly in regions with repetitive sequences or structural

complexities. This limitation is consistent with findings from other studies using short-read data, such as the African human pangenome study (Sherman et al., 2019 (<https://doi.org/10.1038/s41588-018-0273-y>)), which reported similar challenges in assigning contigs to chromosomes

3. Line 174-175: Are the matches to a particular protein domain rather than specific proteins?

Authors' response: Thank you for your insightful comment. You are correct that the observed matches may indeed reflect shared protein domains rather than specific protein identities. In our analysis, we observed instances where multiple transcripts mapped to different regions within a protein. Conversely, we also noted cases where a particular region of a protein consistently matched with multiple query transcripts. This pattern suggests that the transcript regions matching the same protein region may share a common domain.

However, it is important to note that our current analysis primarily focuses on sequence alignments and does not explicitly investigate protein domain annotations. A more comprehensive approach, involving the mapping of identified protein regions to known protein domains using tools like InterProScan or Pfam, would provide deeper insights into the functional implications of these sequence similarities.

4. Line 229-234 and 334-342: The Asian pangenome paper (Dai et al., 2023) showed that about 10% of NRNS were due to introgression from other species. To explain the NRNS finding for *Bos taurus*, rather than acquiring the sequences and then losing them through artificial selection, isn't a more parsimonious (or at least another) possibility that during their evolution the European *Bos taurus* cattle were never exposed to individuals from those *Bos* species and admixture due to inter-species introgression (for the highlighted species) occurred only in the *Bos indicus* lineage?

Authors' response: Thanks for your comment. We were not proposing that *Bos taurus* gained and then lost NRNS. Our hypothesis is that the last common ancestor of taurine and indicine cattle shared NRNS but many were subsequently lost in *Bos taurus* due to natural selection. We do agree that your explanation is also possible and have added it to the discussion. (Line: 368-370)

“This observation suggests a potential loss of ancestral NRNS sequences within the *Bos taurus* lineage, likely due to artificial selection pressures during domestication³⁹. Alternatively, Dai et al., (2023) has shown that about 10% of NRNS in *Bos indicus* were due to introgression from other species. These introgressions would have occurred after the divergence of the *Bos taurus* and *Bos indicus* lineages.”

5. Line 262: How many of the 188,993 novel SNPs were in placed vs. unplaced contigs? Are the ones in unplaced contigs also potentially spurious, especially if they are unplaced because they map to multiple regions of the genome?

Authors' response: Thank you for your thoughtful comment. Out of the 9,825 contigs containing the 188,993 novel SNPs, 35,099 SNPs were identified in placed contigs (1,850 contigs), while 153,894 SNPs were found in unplaced contigs. (Line: 291-293)

It is important to note that only a minor proportion of unplaced contigs are unplaced due to mapping to multiple regions of the genome, which might raise concerns about potential spuriousness. The majority of unplaced contigs remain unplaced because they lack sufficient anchor or split-read support. In this context, anchor or split-read support refers to cases where one read maps to the reference genome, and another read maps to a novel insertion, which is required for a contig to be considered placed.

6. Line 472: Why is a more relaxed sequence identity threshold being used within breed (90%) than between breeds (95%)? This seems backwards.

Authors' response: Thank you for your observation. The difference in sequence identity thresholds reflects the specific objectives and nature of the datasets being analyzed in each scenario.

When comparing between species or with Sahiwal and Tharparkar assembly, we aimed to identify identical or near-identical sequences. In these cases, the subjects were high-quality reference genome assemblies, and the queries were highly accurate NRNS sequences assembled using multiple samples. This high level of accuracy allowed for the use of a stricter identity threshold (95%) to capture highly conserved sequences. This approach is consistent with previous studies, such as Wong et al., 2018 (<https://doi.org/10.1038/s41467-018-05513-w>), which employed similar parameters to detect identical sequences in closely related species.

Within-breed comparisons, however, involved mapping contigs assembled from single samples, often derived from low-depth sequencing reads. This can introduce greater variability and potential errors. To account for this inherent variability and to ensure the identification of potentially relevant matches even in the presence of minor sequence differences, a more relaxed sequence identity threshold of 90% was adopted. This approach aligns with findings from other large-scale genotyping studies, such as the African Pangenome by Sherman et al., 2019 (<https://doi.org/10.1038/s41588-018-0273-y>), which also used a 90% sequence identity threshold for within-population comparisons.

7. Line 579: Where is the VCF file deposited?

Authors' response: Thank you for the suggestion. VCF file are provided as Supplementary Data 2. (Line: 167)

8. Figure 3: What does black and gray on the ideograms represent? Are the chromosomes oriented with centromere to the left? About half of the Brahman reference sequences are backwards compared to ARS-UCD1.2, which are oriented with centromeric end as 0 Mb. Has this been taken into account?

Authors' response: Thank you for your observation. Gray represents the full chromosomes, while black represents gene distribution on the chromosomes. Centromeric regions have not been taken into account. The orientation of the chromosomes is defined by the reference assembly, we have not changed the display due to the random orientation of the reference chromosomes.

9. Supplementary Tables: Recommend sorting all tables by breed.

Authors' response: Thank you for your suggestion. All tables have now been sorted by breed.

10. Supplementary Table 3: The “Seq after Lineage” and “Total Seq” columns are identical. Should they be? If so, delete one.

Authors' response: Thank you for your suggestion. The 'Seq after Lineage' column has been deleted as it was identical to the 'Total Seq' column.

11. Supplementary Table 4: The layout of this table is confusing. Suggest splitting or reformatting. What does “Not hit” mean?

Authors' response: Thank you for the suggestions. It was reformatted correctly for better clarity and renamed as Supplementary Table 5. 'No hit' indicates that the sequence did not match the Bovidae family and was further aligned with the Chordata NR database.

Minor

12. Line 57: Use *Bos taurus indicus* throughout or add *Bos indicus* to the known as clause here.

Authors' response: Thank you for the suggestion. We have incorporated the changes as recommended and have referred to *Bos taurus indicus* as *Bos indicus* in the manuscript. (Line: 57)

13. Line 89: What does this mean: “mapping additional population genomes.” Recommend rewording for clarity.

Authors' response: Thank you for the suggestion. We have reworded the sentence for more clarity.

“We validated the constructed pangenome by aligning sequencing data from genomes of additional individuals representing diverse populations. This analysis demonstrated the pangenome's superiority over the reference genome in terms of read mapping efficiency and its ability to capture novel SNPs.” (Line: 89-92)

14. Line 110: In Supplementary Table 2 the number of PE reads is 21,893 million. Delete “2 x”

Authors' response: Thank you for the observation. We have removed '2x'.

15. Line 111 and throughout: Insert the version of the Brahman reference used. Based on the code for the PanGA pipeline it was UOA_Brahman_1 (GCF_003369695.1).

Authors' response: Thank you for your suggestion. We have included the reference version (GCF_003369695.1) at the beginning of each paragraph where applicable. If multiple mentions occur within the same paragraph, the version is stated only at the beginning for clarity.

16. Line 114: Reword to clarify that the de novo assemblies were performed only for the unaligned reads.

Authors' response: Thank you for the suggestion. We have reworded the text to indicate that the *de novo* assemblies were performed “using their corresponding unaligned reads”. (Line: 118)

17. Line 116: Reword sentence. I initially interpreted “across all samples” to mean you found the same 1000 bp contigs in every sample.

Authors' response: Thank you for the suggestion. We have revised the sentence. “When we selected contigs longer than 1000 bp in every sample, we obtained 244,355 contigs for further analysis.” for better readability. (Line: 119-120)

18. Line 171 and Supplementary Figure 2: StringTie is not represented in the figure.

Authors' response: Thank you for the suggestion. However, we have removed Supplementary Figure 2 based on another reviewer's recommendation that the annotation is already fully described in the methods section.

19. Lines 205, 298, 301: Is significant being used in a statistical sense. If so, report a P-value.

Authors' response: Thanks for your observation. We have appropriately reworded the term “significant” when not used in a statistical sense.

20. Line 380: Were all samples female?

Authors' response: Yes, all samples are female. This information has been included in the manuscript. (Line: 414)

21. Line 384-386: What library preparation methods were used? Barcoding? Same chemistry for both INRP000053 and INRP000159?

Authors' response: We thank the reviewer for pointing out this important oversight. Genomic DNA for both INRP000053 and INRP000159 was prepared for sequencing using the KAPA HyperPlus Kit (Roche, #07962428001). Libraries were barcoded with unique dual-index (UDI) adapters from IDT for Illumina TruSeq. The libraries were quantified using a Qubit 4.0 Fluorometer (Thermo Fisher Scientific, #Q33238) with the Qubit dsDNA HS Assay Kit (Thermo Fisher Scientific, #Q32851). Quality assessment was performed using the TapeStation 4150 (Agilent). Sequencing for both samples was conducted on an Illumina NovaSeq 6000 platform using paired-end 150 bp chemistry. (Line: 417-424)

22. Line 443: Add a citation for CD-HIT.

Authors' response: Thank you for your suggestion, we added the citation.

23. Line 454: Add a citation for ncbi tax2lin

Authors' response: Thank you for your suggestion, we added the citation.

24. Line 550: *Bos taurus* is a subspecies not sister species.

Authors' response: Thank you for the correction. We have revised the text and removed the word sister from the text and now written as "To establish the evolutionary connection between the identified NRNS in our study and species of the Bos genus, we downloaded the reference genomes..." (Line: 597-600)

25. Table 1: Add a footnote listing the accession numbers for the Tharparkar and Sahiwal assemblies.

Authors' response: Thank you for the suggestion. We have added a footnote listing the accession numbers for the Tharparkar and Sahiwal assemblies as recommended.

Reviewer #2 (Remarks to the Author):

Azam et al. present a compelling manuscript analysing nonreference insertions across a large set of diverse Indian cattle breeds, with a detailed analysis on possible functional consequences of such sequence. The results and conclusions are especially important to those in the cattle field but are of general interest to all species where generating long read assemblies for graph pangenomes is as feasible. I find the manuscript overall extremely convincing, although I have several suggestions to address.

Major comments:

After a quick analysis of the NRNS with TRF, almost 42% of the sequences contain tandem repeats (including 2823 out of 13065 with scores>100). These sequences are especially challenging to map, so I think it would be worth analysing how many of these tandem-repeat containing NRNS were amongst the unplaceable NRNS. Interestingly, in the Zhou et al. VCF, only 2259 out of 22324 were TRs with scores>100, so your set of NRNS seems to be capturing much more tandem-repeat variation than previous short read SV pangenomes. Maybe this is a strength of PanGA compared to earlier approaches and could be emphasised?

Authors' response: Thank you for your encouraging feedback on our tool. Our data does show that TRs are a little harder to map. Out of 2,823 contigs, 328 are placed contigs, including 212 single-end and 116 paired-end contigs. The remaining 2,495 contigs are unplaced. This accounts for only 11.61% of the contigs being placed, while a total of 17.61% of NRNS are mapped. There are many differences between the pipelines, so it is difficult to pinpoint why we were able to identify more TRs. (Line: 181-187; 541-543)

Removing contaminants based on databases may be a useful step if you are uncertain about the purity of the sample, but non reference sequence frequently is identified as "contaminants" due to recurring reference-bias (see Rhie et al. <https://www.nature.com/articles/s41586-023-06457-y> in the section: "Contamination of genomic databases"). I find nearly 20% of your NRNS showing up

as “contaminants” as high, but may seem reasonable to you based on the quality of your sample preparation?

Authors’ response: We would like to clarify that a high percentage of reads in each sample were mapped to the reference genome. In fact, the overall alignment percentage to the reference genome ranged from a minimum of 95.86% to a maximum of 99.24% (Line: 115). When averaged across all samples, 98.59% of the data were aligned. We extracted the unaligned reads, which represent an average of 1.41% of the total reads. On average, 20% of these assembled contigs represent approximately 0.3% of the original data. This indicates that only a very small percentage of the data exhibited potential contamination. The identified contaminants are consistent with what would be expected in bovine samples, such as known parasites (e.g., *Theileria annulata* (24.72%), *Theileria orientalis* (7.87%), *Babesia bigemina* (1.22%)) and majority of environmental/uncultured contaminants. (Line: 124-127; Supplementary Table 4)

Given the large improvement in overlapping NRNS against the Tharparkar and Sahiwal assemblies compared to the short read SV VCFs, I would be interested in seeing a repeat of that analysis against e.g., the Hainan indicine assembly from Xia et al. (<https://link.springer.com/article/10.1186/s13059-023-03052-2>). This would help resolve if it is divergence/evolution related (if a low overlap of NRNS with the Hainan assembly) or VCF vs assembly related (if the overlap level is closer to the Tharparkar/Sahiwal assemblies). Especially given Tharparkar and Sahiwal make up around 40% of your samples, I think comparing to an evolutionary-similar but not identical breed is important for this analysis.

Authors’ response: Thank you for this insightful suggestion. We have followed your recommendation and analyzed the overlap of NRNS contigs with the Hainan indicine assembly from Xia et al. (2023). Contrary to our initial expectation, we found a higher number of NRNS contigs aligning to the Hainan assembly compared to the Tharparkar and Sahiwal assemblies.

We have included these results in Table 1 and revised the manuscript. (Line:145-147; 508-509)

This finding suggests that the observed difference in NRNS overlap between short read SV VCFs and assemblies may be more closely related to the data representation (VCF vs. assembly) rather than being primarily driven by evolutionary divergence between the breeds. However, we have not examined this question extensively enough to include in the text.

I think some of the “pangenome” methods would benefit from slightly clearer descriptions when first mentioned. I was unclear what “PanBase” referred to while reading the results and only understood after reading the methods later on. Also when you say “augmenting” the reference, does this mean you concatenated the files together (effectively more unplaced contigs), or is this a graph-style augmentation (like done by vg augment)? Some clarification here would be useful. I’m also somewhat uncertain on how augmenting the linear reference with NRNS would lead to novel variant calls on the sequence that was always present in the linear reference, but maybe a decrease in mismapped reads would improve GATK’s ability to call variants.

Authors' response: Thank you for your valuable feedback. We agree that clearer descriptions of the pangenome methods would enhance the manuscript's readability. We have revised the manuscript to explicitly define "panbase" as the original Brahman reference genome (GCF_003369695.1) without any additional sequences (Line: 620-621). Pangenome sequences were created by concatenating the Brahman reference genome file with the NRNS contig sequences file. We have removed the term "augmenting" from the methods section for better clarity. (Line: 611-613)

Yes, we also concluded that a decrease in mismapped reads improves GATK's ability to call variants. Similar results demonstrating improved variant calling accuracy with pangenome based analyses have been reported in the goat pangenome study (Li et. al., 2019). (Line: 383)

Minor comments:

There are many spacing issues (double spaces between words, no spacing after periods, inconsistent spacing before references, etc) that could be fixed.

Authors' response: Thank you for pointing out the spacing issues. We have carefully reviewed the entire manuscript and corrected all instances of double spaces, missing spaces after periods, and inconsistent spacing before references. We have also ensured consistent spacing throughout the document.

- Remove "staggering" and "impressive" adjectives

Authors' response: Thank you for your suggestion. The adjectives "staggering" and "impressive" have been removed as recommended.

- It is not obvious until the discussion that PanGA was created for this manuscript, so maybe clarify that you use "our PanGA pipeline" rather than "the PanGA pipeline".

Authors' response: Thank you for your suggestion. We have clarified this in the manuscript by using "our PanGA pipeline" at the beginning of the Results section for better clarity. (Line: 123)

- It may be a coincidence, but Ladakhi are the smallest sample size (n=2) and Gir are the largest (n=16), so is observing the fewest/most breed-specific NRNS respectively just a consequence of sample size? Red Sindhi seems to be the main outlier to this, as it has a lot of breed-specific NRNS but only n=6.

Author's response: We acknowledge this limitation. The small sample size in some breeds, such as Ladakhi, may not be sufficient to accurately estimate the true number of breed-specific NRNS. Similarly, the high number of breed-specific NRNS observed in Gir could be partly attributed to its larger sample size. Furthermore, Red Sindhi, with n=6, also exhibits a high number of breed-specific NRNS (696), suggesting that sample size alone may not fully explain the observed

patterns. We emphasize that a larger sample size for each breed would be necessary to definitively characterize the breed-specific NRNS patterns and draw more robust conclusions. (Line: 160-162)

- GO enrichment is easy to over-interpret, but I would be curious to see if there is a difference in GO terms for NRNS-genes found in only a few samples versus many (i.e. cloud- vs core-like genes)

Authors' response: Thank you for this insightful suggestion. We analyzed the GO terms for genes of NRNS found in only a few samples versus many (i.e., cloud- vs. core-like genes), as detailed in the Methods and Results sections. Results are also summarized in Supplementary Figure 3. (Line: 216-222; 574-579)

“GO enrichment for genes from cloud-like and core-like NRNS contigs revealed distinct patterns. Genes associated with core-like NRNS contigs showed significant enrichment for GO terms related to *MHC class II, antigen processing and presentation via MHC class II*, and *MHC protein complex assembly*. This suggests that core-like NRNS genes may have important roles in immune functions. In contrast, genes from cloud-like NRNS contigs were significantly enriched for GO terms related to *cellular component morphogenesis*, indicating potential roles in cell structure and development.”

- Clarify if “23.5% were long interspersed nuclear elements” refers to 23.5% of 39.6% (i.e. 9.3% of the total NRNS) or if it is 23.5% of the total NRNS.

Authors' response: Thank you for your observation. Actually, it is 23.5% of the total NRNS, so we have reworded the sentence to clarify this for better understanding. (Line: 242)

- “identifying identical sequences” the methods describe 95% similarity as the threshold, not identical.

Authors' response: Thank you for your observation. We have reworded the sentence to say "identifying identical or nearly identical sequences" which accommodates sequences with 100% similarity as well as those with similarity close to it (in this case, >95%). (Line: 250)

- “Exotic cattle” seems to refer to *bos taurus taurus*, is this correct? , but I was expecting this to be something uncommon rather than the cattle reference genome.

Authors' response: Sorry for the confusion. In India and other countries where *Bos indicus* is the native subspecies of cattle, *Bos taurus* are considered exotic. (Line: 251)

- “Additionally, research by 31 supports...” Reference 31 has no RNA-seq component, so maybe this was referring to a different Leonard et al (<https://link.springer.com/article/10.1186/s13059-023-02969-y>) that did analyse RNA-seq and non-reference sequence?

Authors' response: Thank you for pointing this out. We appreciate your observation. We have updated the reference to the Leonard et al. study you suggested. (Reference 31)

- The GO term discussion paragraph is interesting, but ultimately quite long and purely speculative based on references to other papers. Perhaps it can be shortened or only keep a few examples to keep the focus on what the novel results in this manuscript are.

..to correctly cite the Leonard et al. article that analyzed RNA-seq and non-reference sequences.

Authors' response: Thank you for your suggestion. We have shortened the GO term discussion paragraph. Additionally, we have ensured that the Leonard et al. article, which analyzed RNA-seq and non-reference sequences, is cited correctly.

- It is likely beyond the scope of what should be done, but I wonder if the NRNS primarily assembled from short reads where one of the read-pairs mapped to the reference had a higher chromosomal placement rate. This would effectively be some type of “flanking sequence”, so would make sense for higher placement rates compared to completely unmapped reads.

Authors' response: Thank you for your insightful suggestion. I completely agree with your observation. However, if we assembled contigs solely from reads where one of the read-pairs mapped to the reference, it would primarily produce flanking sequences at both ends of novel insertions. The method we have used, which is consistent with approaches applied in other studies (e.g., Sherman et al., 2019 (10.1038/s41588-018-0273-y)), assembles not only the flanking sequences but the entire contig. Furthermore, while placing the contigs onto the reference genome, we specifically look for anchoring reads, where one reads maps to the end of the novel contig and the other maps onto the reference. This ensures that the contigs are accurately placed when sufficient anchoring information is available.

- Figure 1. Drop the n=68 as that is dependent on the input, not the pipeline. Add n=68 to the left part and ideally the n= for each breed.

Authors' response: Thank you for the suggestion. We have modified the Figure 1 as recommended. The "n=68" has been added to the left part of the figure, and "n=" has been included for each breed.

- Fig 2. Consider if this should be a Sup Figure. PanGA is a useful contribution to the field, but the primary output of this paper is not the pipeline itself (to my understanding).

Authors' response: Thank you for the suggestion. We have moved Figure 2 to the supplementary section, as recommended, and renamed it Supplementary Figure 1.

- Figure 4A: what does BP/CC/MF refer to? Please describe in the caption.

Authors' response: Thank you for the suggestion. We have added a description of BP (Biological Process), CC (Cellular Component), and MF (Molecular Function) to the figure legend and renamed it as Figure 3A.

- Figure 5. Please create this as an upsetplot similar to Supplementary Figure 1, as a 5-way Venn diagram is not easy to interpret.

Authors' response: Thank you for the suggestion. We personally find the 5-way Venn diagram a little easier to interpret than the upset plots. To accommodate all readers, we have included the upset plots as Supplementary figures 4 and Supplementary figures 5.

- Table 2 and 3. These could be combined into one table. Private-only is only in Table 3, but the only reference to private-only is related to Table 2? Can move the "(of 68)" to the caption rather than listing in every row.

Authors' response: Thank you for the suggestion. We have merged Tables 2 and 3 into a single Table 2. Additionally, "(of 68)" has been moved to the caption instead of listing it in every row.

- Sup Fig 2 is not particularly clear compared to Fig 1 where you provide information on which steps are tools/output/filtering. The annotation is already fully described in the methods, so I would suggest cutting this supplementary figure.

Authors' response: Thank you for the suggestion. We removed Supplementary Figure 2.

- Sup Fig 3 is effectively just a less detailed version of Sup Table 5, which is already referenced in the previous sentence, so I would suggest cutting.

Authors' response: Thank you for the suggestion. We have removed Supplementary Figure 3.

Reviewer #1 (Remarks to the Author):

The authors prepared a careful response to the initial review of their manuscript. The additional analyses undertaken and additions made to the manuscript appropriately address all the major concerns identified. Readability is also improved by the authors' attention to the minor corrections suggested. The results reported are an important contribution to our understanding of genetic diversity of worldwide cattle breeds.

Authors' response: We sincerely thank Reviewer #1 for their positive assessment of our revised manuscript.

Minor

Supplementary Figure 3. The diamonds in the GO plot all appear to be the same size and are the same color. The figure legend suggests variability in size and color is possible. And is it expected in A that they have the same gene ratio? Please double check that this figure has been generated correctly.

Authors' response: Thank you for the reviewer's careful attention to detail. We acknowledge the reviewer's observation regarding the uniformity of the diamonds in the Supplementary Figure 3 Biological Process (BP) graph.

To clarify, we have now updated Supplementary Figure 3 to include the complete set of Gene Ontology (GO) enrichment results, encompassing Biological Process (BP), Cellular Component (CC), and Molecular Function (MF) graphs. This full figure clearly demonstrates the variability in diamond size and color across the different GO categories, as described in the figure legend.

Regarding the uniformity observed in the BP graph alone, we have found that the enriched terms associated with biological processes are limited in number and primarily converge on a few common GO terms. This results in similar gene ratios and, consequently, the uniform appearance of the diamonds in the isolated BP graph. We believe that presenting the complete figure will provide a more comprehensive and accurate representation of our GO enrichment analysis.

Reviewer #2 (Remarks to the Author):

The revised manuscript is largely improved and all of my comments have been satisfactorily addressed.

Authors' response: We sincerely thank them for their time and constructive feedback.